# Oxidative Stress in Sepsis: A Focus on Cardiac Pathology

**DOI:** 10.3390/ijms25052912

**Published:** 2024-03-02

**Authors:** Giuseppe Bertozzi, Michela Ferrara, Aldo Di Fazio, Aniello Maiese, Giuseppe Delogu, Nicola Di Fazio, Vittoria Tortorella, Raffaele La Russa, Vittorio Fineschi

**Affiliations:** 1SIC Medicina Legale, Via Potito Petrone, 85100 Potenza, Italy; gius.brt@gmail.com (G.B.); michela.ferrara@uniroma1.it (M.F.); aldodifaziomedicolegale@gmail.com (A.D.F.); 2Department of Anatomical, Histological, Forensic and Orthopaedic Sciences, Sapienza University of Rome, Viale Regina Elena 336, 00185 Rome, Italy; giuseppe.delogu@uniroma1.it (G.D.); nicola.difazio@uniroma1.it (N.D.F.); vittoria.tortorella@uniroma1.it (V.T.); 3Institute of Legal Medicine, Department of Clinical and Experimental Medicine, University of Pisa, Via Roma 55, 56126 Pisa, Italy; aniello.maiese@unipi.it; 4Department of Clinical Medicine, Public Health, Life and Environment Science, University of L’Aquila, 67100 L’Aquila, Italy; raffaele.larussa@univaq.it

**Keywords:** sepsis, death, oxidative stress, heart, NOX-2, NT, iNOS, 8-OHdG

## Abstract

This study aims to analyze post-mortem human cardiac specimens, to verify and evaluate the existence or extent of oxidative stress in subjects whose cause of death has been traced to sepsis, through immunohistological oxidative/nitrosative stress markers. Indeed, in the present study, i-NOS, NOX2, and nitrotyrosine markers were higher expressed in the septic death group when compared to the control group, associated with also a significant increase in 8-OHdG, highlighting the pivotal role of oxidative stress in septic etiopathogenesis. In particular, 70% of cardiomyocyte nuclei from septic death specimens showed positivity for 8-OHdG. Furthermore, intense and massive NOX2-positive myocyte immunoreaction was noticed in the septic group, as nitrotyrosine immunostaining intense reaction was found in the cardiac cells. These results demonstrated a correlation between oxidative and nitrosative stress imbalance and the pathophysiology of cardiac dysfunction documented in cases of sepsis. Therefore, subsequent studies will focus on the expression of oxidative stress markers in other organs and tissues, as well as on the involvement of the intracellular pattern of apoptosis, to better clarify the complex pathogenesis of multi-organ failure, leading to support the rationale for including therapies targeting redox abnormalities in the management of septic patients.

## 1. Introduction

Sepsis is now defined, according to the 2016 Surviving Sepsis Campaign, as life-threatening organ dysfunction caused by a dysregulated host response following an infection [1]. When circulatory and cellular/metabolic dysfunction appear, the condition evolved into a septic shock with higher mortality [2,3]. Despite research, the pathogenesis of sepsis remains unclear [4].

To date, the universally accredited theory regarding the etiopathogenesis of sepsis which sees a pathogenic agent as a spectator of an uncontrolled inflammatory response, rather than the actual insult, has been questioned [5]. This doubt has been supported by numerous studies that have demonstrated the ineffectiveness of the use of anti-inflammatory agents for the treatment/hyperinflammation hypothesis [6].

Other hypotheses have begun to emerge, attributing a crucial role in the pathogenesis of sepsis to biochemical dysfunctions in cellular oxygen consumption [7,8]. For example, studies in which oxygen administration at supraphysiological concentrations did not improve patient outcomes [9], and since mitochondrial O_2_ consumption represents 90% of total body consumption, mitochondrial dysfunction could be called into question to explain the phenomenology of sepsis.

Oxygen metabolism is, indeed, involved in redox reactions which are essential for the physiological function of cells, in balance between oxidants and antioxidants. It suffices to say that oxidants regulate the formation of deoxyribonucleotides [9]. The free radicals, which are generated during these reactions belonging to the reactive oxygen species (ROS), are also involved in the body’s defense reaction against bacterial infections, both directly and indirectly by acting on vascular tone, cell adhesion, and oxygen concentration [10]. ROS and reactive nitrogen species (RNS), involved in the pathogenesis of sepsis include superoxide (O_2_^−^), hydrogen peroxide (H_2_O_2_) and hydroxyl radicals (HO) [11], nitric oxide (NO) and peroxynitrite (ONOO^−^), the formation of which are due to the innate immune system and specifically to the action of neutrophils and macrophages, which cause the oxidative explosion in the first phases of the sepsis process [12].

In the present study, we focus on cardiovascular dysfunction in sepsis. Septic cardiomyopathy is related to various cardiac alterations that include reversible biventricular dilatation, decreased ejection fraction, impaired response to fluid resuscitation, catecholamine stimulation and redistribution of blood flow between organs, and microcirculatory disorders [13].

Furthermore, cardiac dysfunction during sepsis is associated with high mortality and poor prognosis [2,3]. In detail, firstly, the decrease in arterial contractile capacity in response to noradrenergic stimulation due to sepsis, causes an impairment of vascular homeostasis with hypotension and shock [1]. Secondly, the interaction between cardiomyocytes and LPS generates oxidative stress which contributes to the worsening of conditions in a vicious circle, caused by increased expression of caspases and cellular apoptosis [1].

In this context, the present research project aims to: (i) evaluate the existence or extent of oxidative stress in subjects whose cause of death has been traced to sepsis; (ii) demonstrate its role in the etiopathogenesis and/or exacerbation of sepsis; and consequently, (iii) support the rationale for including therapies targeting redox abnormalities in the management of septic patients. Therefore, this first study focused on the the heart.

## 2. Results

The microscopic analysis of the heart preparations showed a different intensity of positive reaction for each immune marker (Figure 1). All measurements were done on the same magnification of image (40×) and by three different examiners. 

Intense immunopositivity was detected for 8-OHdG, present in approximately 70% of cardiomyocyte nuclei in the group of subjects who died from sepsis (A–D). Furthermore, intense and massive immunopositivity of myocytes in the sepsis group was demonstrated with nox2 markers, iNOS and NT. These data are reported in Table 1. 

The Student’s *t* test was used to attribute significance to the sepsis group/control group. A value of *p* < 0.05 was considered statistically significant. In Table 1, the semi-quantitative evaluation of immunopositivity according to a 0–4 scale has been graphically translated as follows: −: no immunoreactivity (0%); +: mild immunopositivity in scattered cells (10%); ++: immunopositivity in up to one third of cells (33%); +++: immunopositivity in up to two-thirds of cells (70%) and ++++: strong immunopositivity in the majority or all cells (100%). In cases of divergent scoring, a third observer decided the final category.

## 3. Discussion

The present study aimed to evaluate a possible increase in oxidative/nitrosative stress (OS/NS) in cardiac samples taken from subjects whose cause of death was attributed to sepsis [14,15]. The most significant experimental data, in this regard, are the intense positivity assessed by immunohistochemical investigation in the samples taken and treated with markers for the expression of iNOS, NOX-2, and nitrotyrosine. This positivity for all three antigens was significantly increased in the hearts of subjects in the sepsis group compared to the control group, composed of individuals whose deaths had been attributed to other causes [16,17]. This reaction is also associated with the increase in 8-OHdG, therefore supporting the study hypothesis that OS/NS is a key factor in cardiac involvement in sepsis-related death [18].

In detail, the substances produced by oxidative metabolism can influence endothelial-vascular function and a directly cardiotoxic function. Regarding the first: endothelial damage results from the combination of excessive ROS/RNS production and inadequate antioxidant systems (decreased SOD and catalase activity, reduced glutathione (GSH) accumulation, and potential vitamin deficiencies)) [19]. On the other hand, during sepsis there is an acute release of multiple inflammatory mediators (TNF-α, IL-6, and IL-1β) which can lead to both tissue and organ damage, even contributing to cell death [20,21,22].

Furthermore, reactive oxygen species (ROS) in the pathogenesis of sepsis include superoxide (O_2_^−^), which is converted by superoxide dismutase (SOD) to hydrogen peroxide (H_2_O_2_) and hydroxyl radicals (HO) [23,24,25]. Similarly, the reactive nitrogen species (RNS) of interest appear to be nitric oxide (NO) and peroxynitrite (ONOO^−^) [26]. Increased NO levels generate H_2_O_2_ in the mitochondria through inhibition of cytochrome c oxidase. In particular, NO exerts its effect by inhibiting the mitochondrial respiratory chain, in particular complexes I and IV, further complicating oxygen metabolism [27,28].

However, ROS and RNS generated also exert their role in host defense, in fact, deficient O_2_ production is associated with reduced bacterial clearance. Furthermore, lipopolysaccharides (LPS) enhance nitric oxide synthase (NOS), together with the activation of nuclear factor kB (NF-kB) and, consequently, the conversion of L-arginine into NO is promoted, which can be combined with O_2_^−^ to form ONOO^−^ which exacerbates antimicrobial action [29]. 

On the other hand, the synthesis of ONOO^−^ during sepsis is a clear expression of tissue damage resulting from the cytotoxic effect of oxidative stress: (i) ONOO^−^ and its derivatives with intracellular localization can oxidize target molecules, through direct or species-mediated action radicals; (ii) ONOO^−^ promotes the formation of nitrogen dioxide (NO_2_), supporting redox reactions with the consequent formation of nitrate species, such as NT; (iii) ONOO^−^ interacts with nucleic acids, giving rise to 8-hydroxydeoxyguanosine [30,31].

In the present study, cardiovascular dysfunction in sepsis was investigated by following precisely the cellular changes in cardiomyocytes caused by oxidative stress, according to recent advantages in the literature on the subject [32,33,34,35,36,37,38]. Therefore, oxidative stress resulting from the imbalance between the production of ROS/NOS and their elimination, is a source of dysfunction underlying many aspects of CVD [39]. According to Gauthier et al. [40], in fact, at the origin of this evolution there would be a complex balance between the formation of ROS and scavenging in cardiac mitochondria, ascertained through a computational model.

Another fundamental element is the relevance of mitochondrial dyshomeostasis during critical cardiac disease, exemplified by the role played by mitochondria in the damage induced by the ischemia/reperfusion process [41]. Mitochondria generate energy in the form of adenosine triphosphate (ATP) through oxidative phosphorylation (OXPHOS) which causes a 1% to 4% reduction in O_2_ [42]. In cardiac myocytes, mitochondria constitute approximately 30% of the cell volume [43]. Therefore, when certain conditions occur such as hyperinflammation in sepsis, ischemia/reperfusion, and the consequent metabolic reconversion towards aerobic glycolysis can reduce the capacity of the mitochondrial respiratory chain, resulting in cellular stress. The result is an increase in intracellular calcium, the formation of ROS, adenine depletion, and an increase in mitochondrial Ca^+2^ [44]. This leads to increased mitochondrial permeability resulting in mitochondrial swelling and rupture, followed by mitochondria-mediated cell necrosis [45,46].

The loss of cardiomyocytes and inability to regenerate them, leads to fibrosis and cardiac dysfunction [47]. In a study conducted by Zou et al. [48], it has been highlighted that deletion of heart-specific DNA-dependent protein kinase (DNA-PKcs) reduces sepsis-mediated cardiac insult through the improvement of mitochondrial metabolism. Specifically, in cardiomyocytes from DNA-PKcs knockout animal models, the inhibition of the secretion of proinflammatory cytokines restores the glucose metabolic dysfunction caused by LPS. Furthermore, the deletion of DNA-PKcs could overcome the reduced activities of mitochondrial complexes I-III following LPS exposure. In conclusion, the results of these studies demonstrate that deletion of DNA-PKcs in cardiomyocytes post-LPS exposure supports mitochondrial metabolism and respiration, i.e., exerting a cardioprotective effect in cases of myocardial involvement during sepsis [48].

Moreover, septic myocardiopathy appears to be mediated by ferroptosis: LPS promotes the expression of NCOA4 (nuclear receptor coactivator 4), which in turn supports the degradation of ferritin via the Fenton reaction resulting in a higher level of cytoplasmic Fe^2+^, expression of siderofexin receptors on the mitochondrial membrane with consequent increase in the concentration of ferrous ions in the mitochondria which favors ROS generation [49,50]. This in association with the loss of oxidants during sepsis, already explained in the previous paragraphs, together with lipid peroxidation determines ferroptosis [51,52,53]. In their study, Xiao et al. [54] have explored the effects of Fer-1 on sepsis-induced cardiac dysfunction and the underlying mechanism, discovering that Fer-1 improved cardiac systolic function and alleviated cardiac injury through the decrease in intracellular iron accumulation and cardiac and serum inflammatory cytokines levels.

OS/NS also determines the deletion of mitochondrial DNA (mtDNA) of 4977 bp and represents a typical age-related cardiac condition, so much so that it is estimated to be 5 to 15 times higher in people over 40 years of age compared to younger individuals [55,56]. In parallel, 8-oxo-7,8-dihydro-2′deoxyguanosine levels in cardiac mtDNA are found to be inversely proportional to longevity in several mammalian species [57].

In this regard, indeed, the present study demonstrated that the selected markers showed a positive reaction in all the samples examined compared to the control case. In detail, the inducible NOS (iNOS/NOS2) is the enzyme responsible for the production of NO in sepsis in response to endotoxins, cytokines, and other mediators [58,59]. iNOS is, in fact, inactive under physiological conditions and is not constitutively expressed in cells. The increased NO production was correlated with the establishment of hemodynamic changes such as arterial hypotension and increased cardiac work, favoring the onset of septic-induced cardiomyopathy.

Furthermore, LPS stimulation in leukocytes leads to an increase in ROS production through the involvement of the membrane subunit of the NADPH oxidase gp91phox (NOX-2) of phagocytes [60], which, when activated, determines the translocation of its cytosolic components going to determine the constitution of the functional NADPH oxidase complex [61]. The latter, together with cyclooxygenase and the electron transfer chain in the mitochondria, are the cause of O_2_ production [62].

Other studies, on the other hand, have demonstrated that the gp91phox subunit of NADPH oxidase would have a further effect in the induction of cardiac depression resulting from endotoxemia, mediated by the expression of TNF-α, which in turn exerts a negative inotropic effect [63,64].

Furthermore, TNF activation opens the way to two signaling pathways that result in cell death: apoptosis and necroptosis [65,66]. Necroptosis induces a loss of cardiomyocytes, which in turn, as previously described, determines replacement phenomena and cardiac damage in subjects in septic conditions [67,68,69], as summarized and exemplified in Figure 2.

A further elucidation that deserves the study of the samples of subjects who died from SARS-CoV-2 infection, showed an intense positivity especially for the oxidative stress products (8-OHdG and NT), higher than the other preparations (Figure 3).

As previously stated [70], it is well known that COVID-19 must be considered a systemic disease since all organs are affected (lungs, heart, kidneys, liver and brain). In fact, infection by the SARS-CoV-2 virus causes acute respiratory disease (in 71% of cases), myocardial insult (in 33% of cases), acute renal failure (in 20% of cases) and alterations in function liver (in 15% of cases) [71]. Among these, the greatest mortality is related to cardiac involvement [72,73,74]. The etiopathogenesis of myocardial damage could be dependent on direct viral damage accompanied and supported by an indirect insult related to the subsequent cytokine storm (proinflammatory agents, such as cytokines and CD8 cytotoxic cells) resulting in an excessive inflammatory response and, as noted in the present study, the effect of exponentially promoting OS/NS [75,76,77].

## 4. Materials and Methods

### 4.1. Case Selection

The selected cases for the present study had died in hospitals with ante-mortem clinical-laboratory signs diagnostic of septic shock. Three negative control (NC) cases dealing with presumed healthy hearts were selected: a 22-year-old male who expired due to head trauma from a shotgun, and two traffic accidents resulting in the immediate death of a 30-year-old and a 44-year-old respectively, were also included. Familiar and personal cardiologic anamneses were nil. No cardiovascular risk factors were present at the moment of death.

The result of the selection is summarized in Table 2.

### 4.2. IHC

All the samples of tissue were processed and from each sample several sections were prepared: one stained with Haematoxylin and Eosin (H&E), another with Masson trichrome straining, and the other sections were used for the immunohistochemical study according to the panel of antibodies summarized in Table 3 [26].

For immunohistochemical study, 4 μm thick paraffin sections were mounted on slides coated with aminopropyltriethoxysilane (Fluka, Buchs, Switzerland). Pretreatment was necessary to achieve membrane permeability and facilitate antigen retrieval. The primary antibody was applied in a 1:10 ratio for 8-OHdG, in a 1:50 ratio for NOX2, i-NOS, in a 1:100 ratio for nitrotyrosine and incubated for 120 min at 20 °C. The detection system used was the LSAB+kit (Dako, Copenhagen, Denmark), a refined avidin-biotin technique in which a biotinylated secondary antibody reacts with several peroxidase-conjugated streptavidin molecules. Sections were stained with Mayer’s haematoxylin, dehydrated, coverslipped and observed in a Leica DM6000 light microscope (Leica, Cambridge, UK). 

For the semi-quantitative analysis, two observers (GB, AM), blinded, evaluated the intensity of the immunopositive expression according to a scale from 0 to 4: 0 = negative, 1 = mild, i.e., in sparse cells, 2 = immunopositivity ≤ one third of the cells, 3 = immunopositivity ≤ one half of the cells and 4 = strong, i.e., >half or all the cells. In case of divergent scores, a third observer (VF) decided the final category.

## 5. Conclusions

The results demonstrate a correlation between oxidative and nitrosative stress imbalance and the pathophysiology of cardiac dysfunction in sepsis [78,79,80]. This could suggest a greater involvement of oxidative stress in the pathogenesis of septic development [81,82,83]. Therefore, subsequent studies will focus on the expression of oxidative stress markers in other organs and tissues, as well as on the involvement of the intracellular pattern of apoptosis. This will better clarify the complex pathogenesis of multi-organ failure, leading to consideration of the potential therapeutic effects of antioxidant agents in routine therapy in these patients.

## Figures and Tables

**Figure 1 ijms-25-02912-f001:**
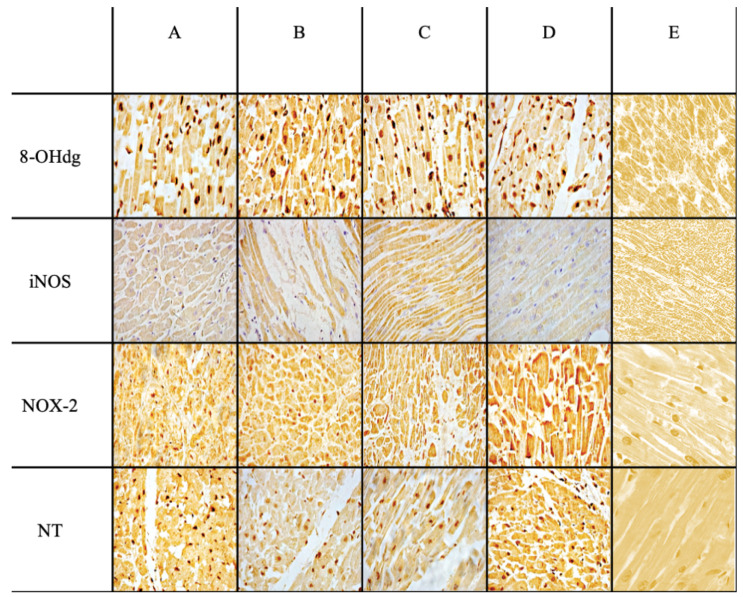
IHC images (40×) from sepsis case examples: the fatal cases are pneumonia sepsis (cases number 1, 3, and 5) and peritonitis (case 10), respectively (original magnification for (**A**–**D**) 40×). Control cases in (**E**) (40×). Immunopositivity for all markers is evident in the cases under examination (**A**–**D**) compared to the control (**E**).

**Figure 2 ijms-25-02912-f002:**
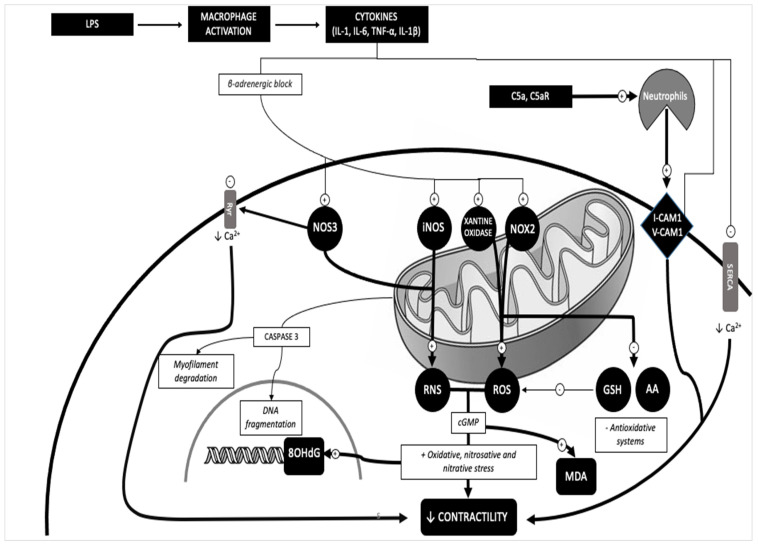
Pathway of cardiovascular dysfunction in sepsis: the expression of iNOS, NOX-2 and NT (evaluated in the present study by immunohistochemical investigations) is significantly increased in the heart of subjects with septic conditions, as is 8-OHdG, supporting the hypothesis that the oxidant/antioxidant imbalance that occurs in OS/NS plays a key role in cardiac cases of sepsis-related death.

**Figure 3 ijms-25-02912-f003:**
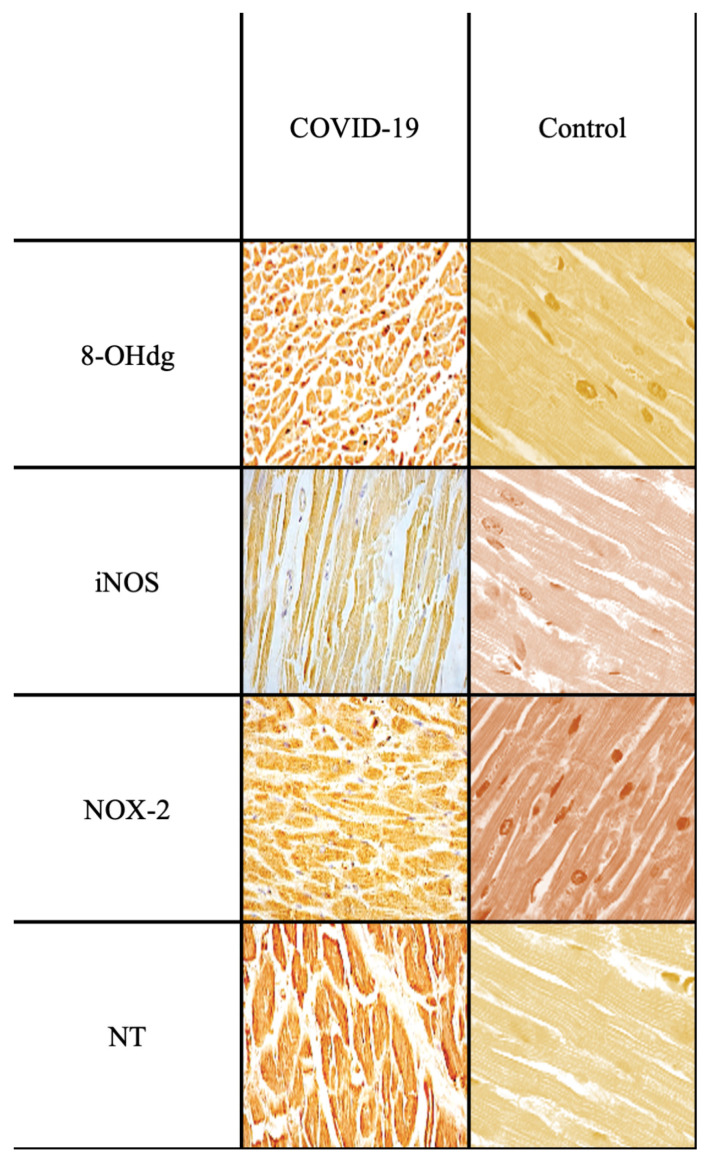
IHC images (40×) from COVID-19 case examples and control specimens (original magnification 40×). iNOS, NOX-2 and NT expressions were found to be significantly increased in septic-death-group hearts with respect to control-heart group. The marker, 8-OHdG also appears significantly increased, supporting the involvement not only of NS but also of OS, even more evident in deaths from SARS-CoV-2 virus infection.

**Table 1 ijms-25-02912-t001:** Statistical analysis of the immunohistochemical findings and gradation of the immunohistochemical reactions.

Ab	Control Group	Sepsis Group	Statistical Value	Sepsis vs. Control
Anti-iNOS	+/−	+++	***	***
Anti-NOX-2	+/−	+++	***	***
Anti-Nitrotyrosine	+	+++	***	***
Anti-8-OHdG	+/−	+++	***	***

Responses to iNOs, NOX-2, Nytrotirosine, -8OHdG, reactions in heart specimens. Results were expressed as mean ± SD. The comparison between groups was conducted using the Student’s *t* test. A value of *p* < 0.05 was considered statistically significant. In table: *** *p* < 0.001.

**Table 2 ijms-25-02912-t002:** Cases and control characteristics.

Case	Sex, Age	Cause of Death Attributed
CASE 1	F, 57 y.o.	Septic shock secondary to pneumonia
CASE 2	M, 62 y.o.	Septic shock secondary to Fournier’s gangrene
CASE 3	M, 68 y.o.	Septic shock secondary to peritonitis due to bowel perforation
CASE 4	F, 75 y.o.	Septic shock secondary to peritonitis due to VAP
CASE 5	M, 27 y.o.	Septic shock secondary to pneumonia
CASE 6	M, 45 y.o.	Septic shock secondary to COVID-19 pneumonia
CASE 7	M, 68 y.o.	Septic shock secondary to cholecystitis
CASE 8	F, 85 y.o.	Septic shock secondary to pneumonia
CASE 9	M, 62 y.o.	Septic shock secondary to bowel perforation
CASE 10	F, 56 y.o.	Septic shock secondary to peritonitis due to enterocolitis
NC	M, 22 y.o	Head trauma from a shotgun
NC	F, 30 y.o.	Traffic accident resulting in the immediate death
NC	M, 44 y.o.	Traffic accident resulting in the immediate death

**Table 3 ijms-25-02912-t003:** Panel of antibodies in study.

NOX-2	gp91-phox, goat polyclonal antibody, sc-5826	Santa Cruz Biotechnology, Inc., Dallas, TX, USA
NT	nitrotyrosine, mouse monoclonal antibody, sc-32757	Santa Cruz Biotechnology, Inc., Dallas, TX, USA
iNOS	nitric oxide sinthases-2, mouse monoclonal antibody, sc-7271	Santa Cruz Biotechnology, Inc., Dallas, TX, USA
8-OHdG	8-hidroxy-2′-deoxyguanosine, mouse monoclonal antibody, N45.1	JaICA, Nikken SEIL Co., Fukuroi, Shizuoka, Japan

## Data Availability

The data presented in this study are available on request from the corresponding author. The data are not publicly available due to investigative secrecy.

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
