# Peer review of "Oxidative Stress in Sepsis: A Focus on Cardiac Pathology"

_ijms, 2024, doi:10.3390/ijms25052912_

Round 1

Reviewer 1 Report

Comments and Suggestions for Authors

The manuscript investigates the role of oxidative and nitrosative stress in heart of patients who died due to sepsis complications.  The purpose of this study is to provide evidence for sepsis-mediated lethality by increased oxidative/nitrosative heart in the context of sepsis in humans.  For this purpose, post-mortem tissues were collected, and heart samples were analyzed by immunostaining for the expression of oxidative DNA damage (8-OH dG), increased oxidative stress (NADPH oxidase 2), nitrosative stress (inducible nitric oxide synthase, 3-nitro tyrosine) and compared to heart specimen from a patient who died due to trauma.

The findings support that there is increased oxidative and nitrosative stress and underlying damage in heart tissues from sepsis patients compared to the heart of the trauma patient.

However, there are some concerns, that need to be addressed to improve this manuscript.

1.    In Fig. 1, please clarify the type of sepsis-mediated mortality is selected for samples A, BC, and D for the immunohistochemistry images. Please also show a representative image of the control (trauma) specimen.

2.    Another suggestion is that the immunostaining for different markers should be shown for the same section and also include the H& E staining as well. 

3.    Also the authors should provide include atleast n=3 for control (trauma samples) for statistical analysis.

4.    In Fig. 3, also again please include representative images from the control specimen.

Comments on the Quality of English Language

Author Response

First of all, we are grateful to the reviewer for the constructive and important comments to improve our paper.

1. In Fig. 1, please clarify the type of sepsis-mediated mortality is selected for samples A, BC, and D for the immunohistochemistry images. Please also show a representative image of the control (trauma) specimen.

We modified the figure as suggested by the reviewer.

2.    Another suggestion is that the immunostaining for different markers should be shown for the same section and also include the H& E staining as well.

As for the possibility of staining an identical section with H&E as opposed to immunohistochemistry, we thank you for the suggestion but feel that it would burden the figure and, as the paper is dedicated to oxidative stress and immunohistochemistry, the H&E images would not provide any additional data. 

3.    Also the authors should provide include at least n=3 for control (trauma samples) for statistical analysis.

Thank you for this suggestion. We did it.

4.    In Fig. 3, also again please include representative images from the control specimen.

Thank you for this suggestion. We did it.

Reviewer 2 Report

Comments and Suggestions for Authors

The manuscript by  Bertozzi G. et al, entitled: Oxidative Stress in Sepsis: a focus on cardiac pathology, presents an interesting study on the influence of the biochemical and immunohistological markers of oxidative/nitrosative stress related to sepsis on the heart tissue.

The analysis was performed on ex vivo heart samples from seven humans that died in hospital from sepsis.

The manuscript is well-structured and presented, and the findings are valuable in terms of correlation between oxidative and nitrosative stress and the pathophysiology of cardiac dysfunction in sepsis.

The authors proposed an ample discussions part, but they should move most of it in the introduction section, for example: lines 135-175, 207-242, and 257-265.

Comments on the Quality of English Language

English language needs only minor editing.

Author Response

The manuscript is well-structured and presented, and the findings are valuable in terms of correlation between oxidative and nitrosative stress and the pathophysiology of cardiac dysfunction in sepsis.

Thank you so much for your kind appreciation about our paper.

The authors proposed an ample discussions part, but they should move most of it in the introduction section, for example: lines 135-175, 207-242, and 257-265.

We did so, we modified the introduction as you suggested.

Reviewer 3 Report

Comments and Suggestions for Authors

Content suggestions:

1.         I would like to kindly ask the Authors to write more about the stsatistical methods used in the study.

2.         Please, include more controls in the study.

Despite the comments, the results of this study will surely be important for the clinical practice improving the management of the patients with sepsis. Therefore, I would like to encourage the Authors to improve the text according to the comments of the reviewers and to resubmit it after the minor revision.

Author Response

Thank you so much for you suggestions to improve our paper.

I would like to kindly ask the Authors to write more about the stsatistical methods used in the study.

We modified the text and the statistical analysis is readable in the text.

Please, include more controls in the study.

We added additional samples of the control cases.

Round 2

Reviewer 1 Report

Comments and Suggestions for Authors

The authors have addressed the reviewer's concerns and have improved the manuscript.  

However, again suggest to include appropriate references in both Introduction (Page 2, Lines 52-65) and Discussion (Page 6, lines 186-189).

Author Response

The authors have addressed the reviewer's concerns and have improved the manuscript.  However, again suggest to include appropriate references in both Introduction (Page 2, Lines 52-65) and Discussion (Page 6, lines 186-189).

Thank you for your suggestion. We added the appropriate references in introduction and discussion too.